# Advance Directives in Portugal: A Qualitative Survey

**DOI:** 10.3390/healthcare12020195

**Published:** 2024-01-13

**Authors:** João Carlos Macedo, Ermelinda Macedo, Rui Nunes

**Affiliations:** 1Nursing School, University of Minho, Campus de Gualtar, 4710-057 Braga, Portugal; emacedo@ese.uminho.pt; 2Health Sciences Research Unit: Nursing (UICISA: E), Nursing School of Coimbra (ESEnfC), Av. Bissaya Barreto, 3046-851 Coimbra, Portugal; 3Research Center for Justice and Governance (JusGov), School of Law, University of Minho, Campus de Gualtar, 4710-057 Braga, Portugal; 4Center of Bioethics, Faculty of Medicine, University of Porto, 4200-319 Porto, Portugal; ruinunes@med.up.pt

**Keywords:** advance directives, living will, healthcare power of attorney, autonomy, end of life, experience, attitude, knowledge

## Abstract

(1) Background: Advance directives (ADs) in Portugal have been legalized since 2012. What has been observed over time, from the few studies carried out, is that despite the positive attitudes in the population, there is a low level of adherence to ADs. To try to understand the reasons for these data, the current study aimed to explore and describe the experiences of the Portuguese population regarding AD. (2) Methods: For this exploratory and descriptive qualitative study, the researchers conducted open (unstructured) interviews with a convenience sample aged over 18 years until data saturation was achieved. (3) Results: A total of fifteen interviews were conducted—eight with women and seven with men. The following four categories emerged from the content analysis of the interviews: (1) AD literacy, (2) AD relevance, (3) AD attitudes, and (4) conditionalities for compiling the ADs. (4) Conclusions: The study pointed out the good receptivity of the participants to the ADs; however, literacy on this subject was low, and identifying the conditionalities in the development of ADs could contribute to improvements in implementation in the population. The data from this study suggest the need to implement measures to increase the literacy of the Portuguese population on ADs and review the legal framework for improving the accessibility of the citizen population. There is also a need to continue researching and obtain more evidence about the ways in which the Portuguese population perceives ADs; thus, in this way, a society can better respond to its citizens’ right to freely exercise their prospective autonomy at the end of their lives.

## 1. Introduction

The exercise of citizen autonomy in healthcare has been developing in recent years [1,2]. Currently, with the great developments in technology, there is an increasing number of therapeutic advances that have allowed us to increase the longevity and quality of life of the population. However, this framework of scientific and technological development is not without ethical problems, particularly in the field of end-of-life care [3,4,5,6,7]. Should all techniques and drugs be used to prolong human life under any circumstance? When should we withdraw or withhold a particular action to avoid medical futility and disrespect for the dignity of the human person? New challenges are posed to average citizens, who, on many occasions, must decide whether to consent or not to act in relation to healthcare. The current paradigm in the provision of healthcare requires respect for the autonomy of a person over the care provided to them, abandoning the paternalistic paradigm that prevailed for centuries [8,9]. In the 1960s, Luis Kutner, a lawyer from Chicago, initiated a new way to respect the autonomy of an individual in healthcare, not only in the present but also in the future [10,11]. Specifically, this refers to an advance directive (AD), a legal instrument that is commonly known as a living will. This instrument, which allows individuals to write their wishes to healthcare professionals, i.e., the care they wish to receive or refuse if they are unable to express their will autonomously in the future, has already been implemented by several countries, although with differing legal formats and characteristics [12,13,14]. Originally, ADs were simply a written document that presented the wishes of a person in terms of the healthcare that they did or did not want to receive, and later, they included greater reflection from the field of bioethics, and the figure of a healthcare power of attorney was also created, where a trusted person who knows the person’s values and desires and would be able to represent them. With evolution and bioethical consensus, it can be said that ADs take two formats: a living will and/or a healthcare power of attorney [15,16,17].

ADs in all legal frameworks in all countries are optional, i.e., a person is not required to prepare such a document. They can only be used to make an advance decision about end-of-life care, i.e., the individuals have the right to exercise prospective autonomy. On the other hand, some studies have revealed the advantages of this instrument in clinical practice, which include ensuring respect for the values and preferences of a patient, increasing the quality of life and comfort at the end of a patient’s life, and avoiding unnecessary medical procedures [18,19,20,21]. The relief of the burden to family members and caregivers in decision-making has also been noted [19,20,22]. Finally, there is an effect on the health professionals, where they are relieved of the anguish of ethical decision-making [23].

In Portugal, the possibility of citizens having this right enshrined had its genesis in the 21st century, more precisely, in 2001, with the Parliament’s ratification of the so-called Convention on Human Rights and Biomedicine of the Council of Europe [24]. This Convention, under Articles 5 and 9 (see below), allowed Portuguese citizens who were unable to communicate to have their previously expressed wishes (in writing or verbally to family members or caregivers) taken into account by healthcare professionals at the appropriate clinical time, as follows:
*Convention on Human Rights and Biomedicine of the Council of Europe**CHAPTER II**Consent**Article 5**General rule**An intervention in the health field may only be carried out after the person concerned has given free and informed consent to it.**This person shall beforehand be given appropriate information as to the purpose and nature of the intervention as well as on its consequences and risks.**The person concerned may freely withdraw consent at any time.**(…)**Article 9**Previously expressed wishes**The previously expressed wishes relating to a medical intervention by a patient who is not, at the time of the intervention, in a state to express his or her wishes shall be considered.*

This was the first step toward the legalization of ADs in the country, but the reflection and bioethical consensus on this matter were advanced in 2006 with the proposal to the Portuguese Parliament launched by the Portuguese Association of Bioethics for the legalization of ADs. It was only in 2012, after social discussion, that law 25/2012 [25], which established the AD as a tool for reinforcing self-determination of a person in healthcare, came into force.

However, more than a decade later, it was found that the Portuguese population’s knowledge and implementation rate of ADs was still very low [26,27,28].

Studies conducted in Portugal on ADs have focused mainly on health professionals [29,30,31,32]. However, few studies have addressed this approach among the general population [27,28,33,34]. The latter, as quantitative research, mainly sought to understand the attitudes and knowledge of the population on the subject. In addition to these data, it is essential to understand how the population perceives ADs. Research on the discourse of the population is essential for us to respond more clearly to the needs they present in this matter.

In view of the above limitations, it was relevant to understand the experiences of people regarding ADs. In this sense, this study was guided by the following research question: how do the Portuguese people deal with the reality of ADs? Therefore, in this study, we intended to explore and describe the experiences of the Portuguese population regarding ADs.

## 2. Materials and Methods

### 2.1. Study Design

This study fit into the qualitative research paradigm, as it was an exploratory, descriptive study that aimed to study the phenomenon in depth, highlighting the meaning that ADs have for individuals [35]. Based on the legal parameters established in Portuguese legislation [25] and the unquestionable respect for human autonomy in the bioethical field, we carried out this study [36].

Using open interviews with the participants, a content analysis was carried out according to Bardin [37], which sought to explore and describe the meaning of ADs for the study population. Furthermore, no software was used to manage the data. In addition, the study followed the guidelines of the Consolidated Criteria for Reporting Qualitative Research (CoreQ) [38].

### 2.2. Instruments, Participants, and Data Collection

It was decided to conduct open interviews to stimulate free thinking and, thus, explore in-depth the discourse of the participants [35].

The interviews followed a previously constructed script for this purpose and included sociodemographic data and the following question related to the subject of the study: would you like to talk to me openly about advance directives?

The study chose an accidental sample, selecting a street in a Portuguese city, going randomly to people’s homes, and inviting everyone over the age of 18 to take part in the study. The interview was pre-tested with a sample of four people. From the evaluation of the pre-test, it was evident that due to a lack of knowledge on the subject, people were unable to speak, which would allow us to carry out the survey. To overcome this problem, it was decided to perform a new pre-test with four other people; however, this time, the participants were given copies of the law 25/2012 [25] and the FAQs [39] so that within a few days, they could read the documentation and then schedule the interview. In this second test, it was verified that after reading the documents, people were able to talk about the subject.

Data collection took place from March to May 2023 by going door-to-door and asking the respondents about their interest in taking part in the research. If they showed interest, the documents mentioned above were handed over and an interview was scheduled for a date and time more convenient for the person and with enough time to read the documents. All the interviews were carried out face-to-face and at a location of their choice. An interview form was used with the socio-demographic data and a guiding question (Appendix A).

### 2.3. Formal and Ethical Procedures

The study was approved by the Ethics Committee of the Faculty of Medicine of the University of Porto (approval protocol n.° 59/CEFMUP/2022). All study procedures followed the recommendations of the Declaration of Helsinki and subsequent updates. Participant consent was obtained through a document in which they were informed in advance of the aims of the study, assured of the anonymity and confidentiality of any data they provided and had to agree to be interviewed by signing their voluntary participation form. They were also informed that they could withdraw from the study at any time without personal harm and that there was no financial or other reward for taking part in the study.

### 2.4. Data Analysis

The interviews were audio-recorded, immediately transcribed, and anonymized with an alphanumeric code (i.e., E1…E15). As these were open interviews, for clarification, the interviewers sometimes asked the participants to increase the depth of the discourse without inducing a response, which basically consisted of interventions such as “Tell me more about that…” or “What do you mean by that…”. For the qualitative analysis of the study, we used the theoretical framework of Bardin [37] and complied with the categorical typology. The data analysis followed three phases. First, the pre-analysis, which consisted of a phase of organizing the material to be analyzed to make it operational and to systematize ideas, corresponded to a period of intuitions that allowed us to guide a precise scheme for the development of the following phases. This was followed by the material exploration/coding phase, which consisted of the systematic transformation of the raw data and their aggregation into units that allowed the description of the characteristics relevant to the content of the text. This was a lengthy stage where the categories and their coding were defined, and the decisions made in the pre-analysis were carried out. Finally, the treatment phase of the results—inference and interpretation—consisted of processing the data to be significant and valid. The relationship between the data obtained and the theoretical foundation provided meaning to the interpretation performed. The treatment and analysis of the data favored an inductive approach, without the definition of a priori categories.

Two researchers (J.C.M. and E.M.) read and reread the interviews shortly after their transcription and independently categorized the discourses, ultimately reaching consensus on the categories and subcategories resulting from the analysis.

## 3. Results

In studies of this nature, the number of participants varies. Thus, the number of interviews ended when data saturation was reached through the data analysis [35,40]. In fact, a total of 15 people were interviewed (eight women and seven men). The youngest was 19 years old, and the oldest was 70 years old. Most of the respondents were married, Catholic, and had higher education. In professional terms, three were retired, two were students, and the remainder had different professional activities, three of which involved health professionals. Of the 15 interviewees, only two had prepared their ADs. The median interview length was 36 min (they ranged from 27 to 72 min). The detailed participant sociodemographic characteristics are provided in Table 1.

Four categories emerged from the content analysis of the interviews: AD literacy, AD relevance, AD attitudes, and conditionalities for compiling ADs. Additionally, 12 subcategories were extracted, as shown in Figure 1.

For the sake of clarity, Table 2 shows the categories, subcategories, and even some excerpts from the participant discourse.

### 3.1. AD Literacy

The category “AD literacy” emerged from the analysis of the interviews because in the speech, there was a lack of knowledge about a subject, difficulty understanding and completing an AD, and poor accessibility to information/training.

#### 3.1.1. AD Unawareness 

Indeed, most participants reported not having knowledge about the subject of ADs. It should be noted that, with the exception of the health professionals involved in the study, the remaining participants had never heard about the subject. This reconfirmed what the researchers found during the pre-test of the interview (see Section 2.2), where they had to send previous documentation to interviewees to be able to formulate some discourse on ADs.

#### 3.1.2. Difficulties with Understanding and Completion

Another aspect that was of relevance was the recording of difficulties in understanding the documentation of a living will and the manifest difficulty in writing an AD. There were participants in the study who reported that the optional document for writing an AD was not understandable and the language was not accessible, and they were unable to write it without some help. There were references to the need to ask a health professional for help in order to develop an AD.

#### 3.1.3. Weak Accessibility to Information/Formation

Another aspect that emerged from the speech was poor accessibility to information/training. In fact, only the health professionals had the opportunity to receive training on the subject. In addition to not having knowledge about the subject, as mentioned in Section 3.1.1 of the results, the remaining participants were also unable to identify access to information on ADs. There was one interviewee who referred to a memory during the initial phase of the process where there was some disclosure in the media and some debate in the community, and another interviewee remembered having seen a dissemination poster in a health unit.

### 3.2. AD Relevance

Another category that emerged from the discourse was AD relevance, and it took two forms: personal and family relevance.

#### 3.2.1. Personal Relevance

A majority of the sample highly valued personal autonomy with regard to end-of-life care. The interviewees pointed to personal wills and the exercise of autonomy as fundamental elements of respect for a person. Thus, ADs assumed personal relevance because, in their words, they are fundamental instruments for respecting a person’s dignity at the end of their life.

#### 3.2.2. Family Relevance

An awareness of the importance of having ADs for one’s family also emerged in the speech. The analyzed data allowed us to show that the interviewees did not want to be weights or burdens at the end of their lives, and they verbalized that having ADs would be elements that would not burden their families when they had to make decisions.

### 3.3. AD Attitudes

From the analysis of the interviews, a set of elements also emerged that allowed us to categorize attitudes toward ADs.

#### 3.3.1. Positive

Most of the interviewees had positive attitudes, considered ADs important, and in some cases, intended to elaborate on ADs and considered it logical to have a law of this nature.

#### 3.3.2. Negative

A negative attitude emerged in this sample, and it showed no advantages in implementing the ADs.

#### 3.3.3. Neutral

Two attitudinal positions also emerged from the interviewees that were configured as neutral. Respondents considered that they ethically accepted the ADs but were indifferent (i.e., neither in favor of them nor against them).

### 3.4. Conditionalities for Compiling ADs

During the interviews, the researchers were also able to find important data relating to the constraints on the development of an AD.

#### 3.4.1. Religious Beliefs

One of the constraints that was mentioned, in this case, as an impediment to developing an AD was religious belief. Participants of the sample, mostly Catholics, revealed that they believed in natural death without interference; therefore, drawing up a living will would interfere with this process. According to one participant, developing an AD did not seem to be in accordance with her religious beliefs or philosophy of life.

#### 3.4.2. Previous Experience with Death and Dying

Another conditioning factor emerged for the elaboration of ADs, and it had to do with previous experience with death and dying. From the speech obtained, we were able to verify that the experience of monitoring or even caring for a family member at the end of their life is a motivator for developing an AD. The course of illness of the family member until death, especially for those with very disabling diseases, was a reason for reflection and constituted an impetus to think about the need to express wishes about end-of-life care.

#### 3.4.3. Death Taboo

In the discourse of the interviewees, the taboo of death also emerged as an impediment to the elaboration of an AD. Indeed, some interviewees expressed their avoidance of talking about the subject and how uncomfortable it was for them. According to them, preparing for the end of life is something they do not think about; therefore, they will not seek to develop an AD.

#### 3.4.4. Bureaucratic Issues

Another constraint also emerged which, according to the participants, was an obstacle to the elaboration of an AD, and it had more to do with the bureaucratic aspects of law enforcement. There was reference to the difficulties in understanding the technical language of the optional model that was placed online for citizens to fill out. Further, the renewal of an AD must be completed in-person after five years, and the participants questioned this period of validity for an AD and the need for renewal to require the person to go to the service with a new document. Participants considered the bureaucratic aspects to be constraints on the elaboration of ADs.

## 4. Discussion

To the best of our knowledge, this was the first study conducted in Portugal to explore and describe the experience of the Portuguese population with regard to ADs. The data found allowed us to develop a picture, although without generalizations, of how Portuguese citizens understood their right to exercise prospective autonomy in end-of-life care.

The results obtained in relation to literacy of the sample on the ADs, which revealed a lack of knowledge on the subject, were consistent with the findings of other studies [20,27,41,42,43,44,45,46,47,48,49]. A systematic review of the literature published from 1994 to 2016, revealed a low level of knowledge about ADs [19]. More recently, another review of literature published from 2018 to 2022 confirmed this fact [50]. As mentioned in the data collection section, it was only after reading the documents provided beforehand that some of the participants became aware of ADs. The Portuguese legislation on ADs dates back to 2012, and in 2017, the Portuguese parliament, realizing the low level of population adherence to ADs, asked the government to promote a publicity campaign [51]. Based on the data obtained thus far on the knowledge and level of adherence to ADs, it appears that the campaign did not have good results or that the dissemination strategies were not effective; therefore, there is still a need to promote the dissemination of ADs among the population [15,52,53].

In addition, only two participants in the sample had ADs, which was in agreement with the low level of adherence that existed at the national level. According to data from the Shared Services of the Ministry of Health, this rate was 0.51% of the population [26]. With the exception of the United States, where approximately one-third of the population has an AD [54,55], in European countries, the prevalence of ADs is lower, between 0.66% and 19% [41,43,56,57]. Another area identified in the context of literacy about ADs had to do with difficulties in understanding the AD documents and the wording of the ADs. These data likely explain the low adherence rates, constituting a barrier for citizens to exercise this right, as studies in other countries have shown [18,58,59]. Additionally, for this item, the participants mentioned the need for help in preparing ADs. Interestingly, this issue of citizens obtaining help with preparing their ADs has been expressed for some years in the United States through the Federal Patient Self-Determination Act of 1990, which forced health institutions to inform citizens of this right and provide counseling for its realization [60,61]. This aspect was particularly relevant, given the low level of literacy of the population in this area, due to both a lack of knowledge and an inability to understand what to write in an AD. It appeared that the need for counseling by health professionals, who are, in principle, more technically qualified, was a measure to be implemented to increase the ratio of ADs [53,62,63,64,65,66,67,68,69,70].

However, in this context, it turned out that there was scarce information and/or training on ADs in the population. It should be noted that within the sample, only health professionals had training on the subject. It appeared that accessibility to information/training was low, and furthermore, the Portuguese government’s publicizing of ADs to the Portuguese population at Parliament’s request [51], as mentioned in the introduction, did not have much effect. There was a need to increase access to training, especially for health professionals, to promote counseling, as mentioned above, and to increase access to community information in this digital age on social networks [71,72], support websites, or applications [73,74], as suggested by some studies. 

Despite these aspects of literacy, the results of the study showed that attitudes toward ADs were mostly positive, which agreed with the findings of other studies [18,27,47,48,49,56,58,75,76,77]. However, there were also negative and even neutral attitudes. It is believed that this negative attitude may be associated with religious beliefs, that neutral positions were reported by younger participants, and that being confused about ADs demonstrated an uncommitted position.

Another interesting finding that emerged from the interviews was the relevance that the participants gave to the ADs. In addition to the positive attitudes, the participants attributed dual relevance to the ADs, as follows: they assumed the right to exercise their autonomy as fundamental in healthcare, and thus, in this way, ADs alleviated the burdens on their families for decision-making, given that these findings were in line with other empirical findings [19,23,54].

There is still much to be added to the findings of this study regarding the conditions for developing an AD. In this regard, religious beliefs likely play a role in hindering the development of ADs in the Portuguese context. Participants claimed that they did not want to interfere with the death process, and they felt that ADs interfered with this natural end-of-life process. Although these data need to be further investigated in terms of research, and bearing in mind that the majority of the sample was Catholic, it could be a notion of defense against natural death and the sacredness of life, which are ideas that are present in documents of the Catholic Church [78]. Curiously, in the United States, the Catholic Church informs and educates the faithful about ADs [79], but in Portugal, there has not been the same level of attention. This may determine this particularity and that it is still a subject that has not been reconciled with faith due to a lack of knowledge. Interestingly, some studies have shown that the more religious a population is, the less likely people are to think about the subject or develop an AD [80,81,82].

Previous experience and contact with death and dying were described as motivating individuals to develop ADs. Indeed, these data were in agreement with other studies that reaffirmed that experiencing the course of a disease in the death of a loved one can lead to introspection, identifying the values and care that one wants to receive or refuse at the end of their life, and thus, they seek to develop an AD [83,84].

In contrast to these results, it appeared to us that death was a barrier to the development of an AD, which was also what emerged in the discourse of some of the participants. Indeed, the disagreement in talking about death and even the denial of it could be assumed to be obstacles to discussing the planning of end-of-life care, as has been evidenced by some studies [76,77,85]. In this regard, there appears to be a need to introduce the themes of death and dying in the social discourse to demystify these subjects, reducing the fear, anxiety, and avoidance that are present when addressing end-of-life issues [86,87,88,89,90,91,92]. In recent years, the need for conversations about serious illness has been discussed and researched, and goals, values, fears, worries, desires, and hopes have been discussed on journeys through illnesses and the human experience of living with a serious illness has been explored, centered on people and their problems [93]. This could be a strategy for promoting end-of-life-care planning and ADs.

Despite the constraints in the elaboration of the ADs, some bureaucratic data have also emerged. The participants claimed that some items that are imposed by the AD law [25] may constitute obstacles in the implementation of ADs in a community and that they have to do with the need for renewal of an AD in-person and the five-year validity period. This aspect is relevant because the law is more than ten years old, and the data from this study point to the need to revisit the law and update it.

## 5. Limitations and Strengths

This qualitative study attempted to explore in-depth the experience of the Portuguese population regarding ADs; however, it had several limitations. First, it was not possible to make generalizations, and the convenience sample obtained may not have been able to encompass all the perceptions of the population regarding the subject under study. On the other hand, this sample had very specific characteristics, for example, more than half of the participants had higher education, which could also be a limiting element and not sufficiently comprehensive of the entire Portuguese population. Despite this, this study had its strengths in that it was the first study to be carried out with these characteristics in Portugal, providing a more concrete, albeit limited, understanding of how the Portuguese population deals with ADs.

Although the data obtained in this study cannot be generalized, the assumption of transferability is implicit in these studies, and it is understood as the possibility of observing the same result in other groups under similar social conditions, given the specificities of the new context [94]. Thus, it is necessary to continue investigating other samples and other types of investigations via focus groups to obtain additional evidence.

## 6. Conclusions

This study shed some light on the way Portuguese people deal with ADs. The data obtained indicated low levels of literacy, positive attitudes, and the special relevance of ADs for the Portuguese population. On the other hand, we identified some constraints that may facilitate or hinder the development of ADs in society.

As a result of this study, we developed the following implications for education: improving training programs for the general population and for health professionals, and it is also important that the curricula of students in health areas can, from the early stages, begin their training on end-of-life-care planning. On the other hand, there are also implications for the policy area because there is a need to change and improve the AD law. Finally, there is also a need to continue researching this topic to obtain additional data and to respond to society’s need more effectively for end-of-life-care planning. In this way, citizens can be given the chance to freely exercise their prospective autonomy.

## Figures and Tables

**Figure 1 healthcare-12-00195-f001:**
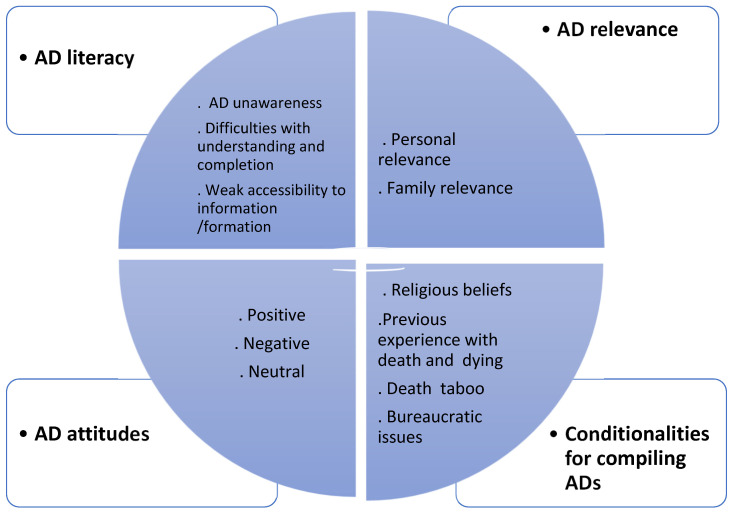
General overview of the categories and subcategories identified from the interview analysis.

**Table 1 healthcare-12-00195-t001:** Participants’ sociodemographic characteristics (*n* = 15).

Variable		*n*
Age	18–28 years old	2
	29–39 years old	2
	40–50 years old	5
	51–61 years old	3
	≥62 years old	3
Gender	Female	8
	Male	7
Marital status	Single	3
	Married/living together	12
	Divorced/separated	1
Religion	Catholic	12
	Agnostic/atheist	3
Level of education	Elementary education	2
	Secondary education	5
	Bachelor’s degree	5
	Master’s degree	3
Occupation	Business manager	1
	Commercial employee	3
	Dental assistant	1
	Nurse	1
	Physician	2
	Retired	3
	Student	2
	Teacher	2
Have a living will	Yes	2
	No	13

**Table 2 healthcare-12-00195-t002:** Categories, subcategories, and context unit (verbatims).

Category	Subcategory	Unit Context (Verbatims)
**AD literacy**	**AD unawareness**	“…for me, it is total ignorance…” (E2) “…I have no knowledge of AD, in fact I learned very recently that it exists, now… what I know is very vague, I would like more information…” (E5)
	**Difficulties with understanding and completion**	“…the ADs, the document, is difficult to understand… and… so I chose to make another one that a friend of mine made easy for me…” (E1) “…the document is not very understandable in my opinion, is it? It is a bit difficult to understand, to understand what’s written there…” (E11) “…no, I do not know how to do it, I would not even know how to fill in the document…” (E8) “…at the moment, if I were to do it now, no, I would not have the capacity, the knowledge to do a living will…” (E4)
	**Weak accessibility to information/formation**	“…there was an initial phase when the document appeared that…, it was somehow publicized through, through the media and perhaps through some debates that appeared on television and after that I have no memory of the subject being discussed any more and perhaps it fell a little into oblivion, at least that is the feeling I have given that I have never seen any more news on the subject…” (E2) “…I learned about the living will during in-service training…” (E12)
**AD relevance**	**Personal relevance**	“…this right to my autonomy, I want to be autonomous and I do not want to be a burden…” (E7) “… it is important for the person to make the decision of what they want to happen, from the moment they can no longer be conscious (…) I have the right to choose…” (E14)
	**Family relevance**	“…and I do not think my children should have to suffer for a long time in a situation where there will be no return, so I thought it was the best thing for me to do and I did it, I made a living will…” (E1) “…and lightness for those around me, so that they do not have to deal with issues that are so difficult, so heavy and that can lead to so much guilt for them, that I do not want that…” (E6)
**AD attitudes**	**Positive**	“…the document makes sense so that I can express my intentions for a certain type of care when I cannot decide for myself…” (E13) “I think it is very good that we’re allowed to do AD because it is important for people to make decisions about what they want to happen in the future, while still being aware.” (E14)
	**Negative**	“…I do not see much advantage in the living will (…) I’m not in favor of it.” (E9)
	**Neutral**	“…ethically I have to accept it…I’m neither for nor against it…” (E12)
**Conditionalities for compiling ADs**	**Religious beliefs**	“In my religious belief, I do not think it is good to do these things (…) to let life run its course, that is how it is… (E12) “I think that natural death is the best thing, there should be no interference, I think that it’s for another plane (…) for another entity, these are things that I keep in mind in what I’ve always believed.” (E9)
	**Previous experience with death** **and dying**	“… and if it experienced this a lot, the death of someone close to me, was from then on that I also got the idea that if something similar happened to me, I’d rather they did not give me the treatment to prolong it…” (E3) “… Seeing his situation and his illness, I fully accept his wishes… and for me I think it is something important and eventually I will even do the TV, because in fact we’re here and we never know tomorrow and anything can happen, accidents, illnesses and so on, and so it’s something that’s up to us to decide, in fact, to make the decision.” (E10)
	**Death taboo**	“…ah the end of life…, we immediately change the conversation, we immediately avoid the conversation…” (E9) “… I do not think about it much either, we talk about when someone dies, but preparing for the end… I’m honest, it is a subject I avoid.” (E15)
	**Bureaucratic issues**	“…it seems to me that the bureaucracy is something that does not help, for example the optional model has very technical language for the citizen to understand… also in terms of renewing the DAV it has to improve, and perhaps it can be done at home,…there are aspects to improve.” (E11) “Perhaps less validity time, and perhaps traveling causes some embarrassment… there would be more places and digitalization would be easier… on the other hand, the model is not easy to fill in, it is very technical…” (E13)

## Data Availability

All data generated or analyzed during this study are included in this article.

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
