# Peer review of "Advance Directives in Portugal: A Qualitative Survey"

_healthcare, 2024, doi:10.3390/healthcare12020195_

Round 1

Reviewer 1 Report

Comments and Suggestions for Authors

As in attached file

Comments on the Quality of English Language

There is a need to look at this area as described in the attached review

Reviewer 2 Report

Comments and Suggestions for Authors

What  they found is normal when you pick a random sample and ask people questions such as "What do you think about death?" Much better would have been to separate those in health care (who may have advocated for ADs) from family members who experienced a difficult care situation in a hospital for a terminally ill person, and then, maybe from those with no background or personal stake in the issue. The push for ADs was an effort for specific interest groups, who then publicized the possibility in the USA. Part of that publicity came through media stories (very popular here) about people "kept alive by machines," possibly against their will. The failures of that effort came because ADs were often not heeded, or bypassed by family who insisted "Do everything possible." From this essay, it sounds as if Portugal had little of these examples, or less private advocacy. To depend on the government to provide information obviously did not work. But the authors did not seem to frame the issue except in terms of "illiteracy," and because they interviewed the random sample, they did not learn as much as they should have.

A curious issue is the way religion in this essay is located as an obstacle to understanding or approving of ADs. Since the Catholic Church long ago pioneered the distinction between ordinary and extraordinary means to preserve life, they set the stage for acceptance of some limits on technology to extend dying. The argument was actually in favor of some degree of "natural death," which is what most ADs support. There is no absolute obligation to preserve life at all costs, in other words. Apparently, the Church did not educate persons about this, and/or people adopted a more stringent view that their religion teaches the opposite. This is very unclear in the paper.

I think the topic is interesting, but only if the research design is altered to as to discover if and how personal connection with the whole issue makes a difference in how people understand what is at stake.

Comments on the Quality of English Language

The authors need an editor to correct or clarify some of their English usage. Mostly, it's ok, then the reader finds prepositions that do not fit, or wordings that are awkward. 

Reviewer 3 Report

Comments and Suggestions for Authors

The article is quite interesting and well-written. It has its merits, but it also has some drawbacks. In order to improve the quality of the text, certain changes should be made:

Line 39 – remove the period/dot.

Line 40 – replace "futility" with "medical futility."

Line 47 – remove the period/dot; in many places in the article, there are two periods at the end of some sentences. They should be removed.

Lines 76-90 – I suggest presenting the content of the articles in a table. This will enhance the clarity of the text.

Line 233 – It is written: "respecting autonomy and dignity at the end of life." Does autonomy differ from dignity, or are they the same? What position has been adopted in this article? I ask this because the discussion around autonomy and dignity in bioethics or medical ethics is very complex. I think that in this specific place, it is sufficient to write "respect for dignity."

Lines 345-350 – I do not believe that religious beliefs must be an obstacle to the development of ADs. Ethical and anthropological assumptions may underlie religious beliefs. There are numerous arguments in the literature that speak against ADs (See: Beauchamp, Tom L., Childress, Principles of Biomedical Ethics, Oxford 2019).

Weaknesses of the article: (a) A small number of study participants. It is therefore unclear whether the studies can be considered representative, (b) Short data collection period (See: line 137).

Round 2

Reviewer 2 Report

Comments and Suggestions for Authors

This rewrite improved the clarity and major points of the original paper. The principle finding is that people in Portugal do not know about what ADs do, what their provisions  and limits are. The legislation was pushed by healthcare professionals, and relied on the ideal of autonomy. But without strong public sentiments, or popular culture's contributions to death awareness, the whole effort remains unsuccessful. The authors make their point, and some of the unhelpful English usage has been carefully corrected. Therefore I believe this new version should be published.